# All Models are Biased, Some are More Transparent about it: Fully Interpretable and Adjustable Model for Mental Disorder Diagnosis

## Abstract

Recent advances in machine learning have enabled AI applications in mental disorder diagnosis, but many methods remain black-box or rely on post-hoc explanations which are not straightforward or actionable for mental health practitioners. Meanwhile, interpretable methods, such as k-nearest neighbors (k-NN) classification, struggle with complex or high-dimensional data. A network-based k-NN model (NN-kNN) combines the interpretability with the predictive power of neural networks. The model prediction can be fully explained in terms of activated features and neighboring cases. We experimented with the model to predict the risks of depression and interviewed practitioners. The feedback of the practitioners emphasized the model's adaptability, integration of clinical expertise, and transparency in the diagnostic process, highlighting its potential to ethically improve the diagnostic precision and confidence of the practitioner.

## 1 Introduction

The booming era of Artificial Intelligence sees a rise of its application to mental health-related issues (Graham et al., 2019) and specifically to mental disorder diagnosis ("diagnosis" for short) (Bzdok & Meyer-Lindenberg, 2018; Chattopadhyay, 2017; Graham et al., 2019; Iyortsuun et al., 2023). Although various AI technologies have achieved high accuracies in diagnosis, they generally lack explanation for their decision making, and the process remains a black box for both mental health practitioners ("practitioners" for short) and clients seeking counseling services ("clients" for short) (Jarvie & Lindén, 2024; Lau et al., 2023). Therefore, there is still much caution and concern about the use of AI for diagnosis (Kerz et al., 2023; Li et al., 2024).

This gives rise to recent trends in using Explainable AI (XAI) technologies for mental health: Practitioners and clients may rely on AI tools "to the extent they can economise on human oversight, monitoring and verification of the system's outputs" (Joyce et al., 2023). Most XAI methods are based on post-hoc explanation methods such as SHAP (Shapley, 1953) and LIME (Ribeiro et al., 2016). However, Rudin (2019) argues that post-hoc explanations are often inadequate for black-box models. The explanations may justify the model's decision after the decision is made, but not how the model reached the decision internally. In the worst cases, post-hoc explanations are excuses for a model's mistakes and offer no opportunity to debug or fine-tune a trained black-box model.

To address the interpretability and adjustability of AI models that facilitate practitioners in diagnosis, we propose to use a recently invented model, a neural network based k-nearest-neighbor algorithm (NN-kNN). As a k-nearest neighbor algorithm, NN-kNN can explain each model decision with activated cases, and each activated case can be attributed to its feature distances with the query. As a neural network, NN-kNN allows end-to-end training for both feature weights and case weights, and is compatible with other neural network methods (Ye et al., 2024).

Our study introduces a novel approach to human-machine interaction drawing on insights and methodology from both AI and psychology. Specifically, we study how NN-kNN facilitates practitioners in aspects beyond traditional diagnosis, including feature exploration, explanation of decisions, past case retrieval, and manual weight adjustment. We conducted a qualitative study, using

interpretative phenomenological methods to capture the insights of the practitioner's experience with the interpretable and adjustable model. Although qualitative research is common in the field of psychology, incorporating this method in AI research offers a novel perspective. This approach enriches existing XAI research and expands the understanding of human interaction with machine learning models, as discussed in Section 4.3.

This article is organized as follows. Section 2 describes the interpretable model we use. Section 3 discusses related work in both the fields of explainable AI and mental health diagnosis. Sections 4 and 5 describe the qualitative study method and findings. Lastly, the article concludes with discussions and future directions.

## 2 NEURAL NETWORK BASED K-NEAREST NEIGHBORS

### 2.1 K-NEAREST NEIGHBORS CLASSIFIERS

K-nearest neighbors classifiers (k-NN) examine the task domains involving cases $C$ in the form of $(x, L_x)$. For each case $x \in C$, $f(x) = < x_1, x_2, ..., x_m >$ is a feature vector describing a client's information (survey answers, medical record information, etc.) and $L_x$ is the diagnosis label associated with the client (risk of depression). The function $f$ might describe a feature extraction method or simply use the surface features of $x$. A naive k-NN calculates the distance between two cases as the sum of distances between corresponding features, using a simple Minkowski distance measure such as the Euclidean distance (Dasarathy, 1991). Better k-NN methods use feature weights in the calculation of distance. Traditional methods use a global feature weighting, while others allow certain sets of cases (or even each individual case) to have their own feature weighting (Manzali et al., 2024; Aha & Goldstone, 1992; Friedman, 1994; Ricci & Avesani, 1995; Marchiori, 2013; Bonzano et al., 1997). Some methods assign case weights, so that certain cases contribute more in the voting of the final prediction (Bicego & Loog, 2016; Aguilera et al., 2019). The case weights may be based on the distance between the case and the query, or it may be a learned parameter of the case. Other methods such as neighborhood components analysis (NCA) and large margin nearest neighbor (LMNN) transform the feature space to extract high-level features before distance calculation (Goldberger et al., 2004; Weinberger & Saul, 2009).

### 2.2 NEURAL NETWORK BASED K-NEAREST NEIGHBORS ALGORITHM

The neural network based k-nearest neighbors algorithm (NN-kNN) implements both feature weights and case weights by having network layers that simulate the behavior of a k-NN:

1. The case layer stores all cases (the training set).

2. The feature extraction layer extracts the features of the query $q$ and each case $x$.

$$f(q) = < q_1, q_2, ..., q_m >, f(x) = < x_1, x_2, ..., x_m > \tag{1}$$

3. The feature distance layer calculates the distance between the corresponding features as

$$\delta_i = \delta_i(q_i, x_i) \geq 0 \tag{2}$$

We choose the squared distance $\delta_i(q_i, x_i) = (q_i - x_i)^2$

4. The case activation layer activates a case $x$ given the query $q$ by

$$case\_activation(x|q) = \sigma(w_{x\delta_1} * \delta_1 + w_{x\delta_2} * \delta_2 + ... + b_x) \tag{3}$$

where $w_{x\delta_i}$ is the weighting of feature $i$ for case $x$ and $b_x \geq 0$ is the default activation for case $x$. $w_{x\delta_i}$ can only be negative because the more different the case from the query, the less activated the case. We choose the sigmoid function $\sigma()$ as the activation function to limit $case\_activation$ within $[0, 1]$.

5. The top $k$ case selection layer is optional. When enabled, it will keep the top $k$ case activations and resets other case activations to 0.

6. The target activation layer takes each case's activation to activate the corresponding class.

$$class\_activation(L|q) = \Sigma_x(w_{(x,L)} * case\_activation(x)) + b_L \tag{4}$$

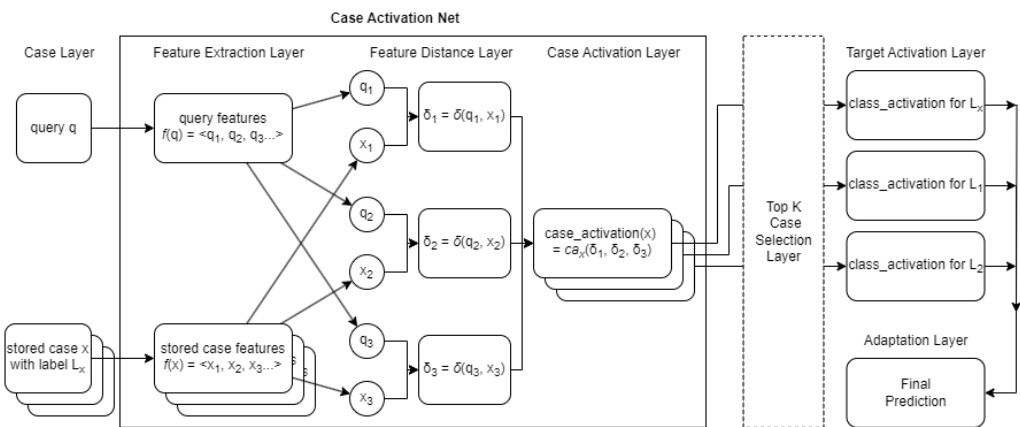

Figure 1: The Model of NN-kNN (Ye et al., 2024)

where $w_{(x,L)}$ is the weight of the case $x$ for the class $L$ and $b_L$ is the bias of the class $L$. $w_{(x,L)}$ is forced to be positive (by using a ReLu) if the case $x$ is of class $L$ and $w_{(x,L)} = 0$ otherwise, because a case should only activate its corresponding class but not other classes.

7. The adaptation layer chooses the label $L$ with the maximum $class\_activation$.

The layers are depicted in Figure 1. In previous experiments comparing NN-kNN with neural networks and state-of-the-art k-NN methods (such as LMNN), NN-kNN achieves equal or less prediction error in classification and regression on multiple datasets (Ye et al., 2024).

NN-kNN has the potential to incorporate more advanced neural architectures. For instance, additional layers can be added in the feature extraction layer to transform the feature space like NCA or LMNN does. Although adding deeper layers to NN-kNN can enhance its analytical capabilities, it comes at the cost of reduced interpretability. Greater analytical power is not always essential, and interpretable models often achieve good enough performance, sometimes even surpassing non-interpretable models (Rudin, 2019; Ye et al., 2024).

## 3 RELATED WORK

### 3.1 EXPLAINABLE AND INTERPRETABLE AI

According to the survey by Joyce et al. (2023), the majority of XAI methods in mental health are based on feature importance methods such as SHAP (Shapley, 1953). Feature importance methods estimate the weights of features by measuring the influence on the model's output after perturbing a feature. Only two methods in the survey are regression-based and interpretable by design. Similarly, most studies in the wider field of XAI are post-hoc methods because they are flexible and easily applicable to different models (Saleem et al., 2022). Post-hoc methods build an interpretable model that mimics the behavior of a black-box model and use this new model as an explanation. However, post-hoc explanations are problematic for high-stakes decision making in mental health applications: (1) post-hoc explanations may not faithfully represent the original model's computations, (2) they may lack the detail necessary to fully understand the black-box model, and (3) they do not permit manual calibration by domain experts (Rudin, 2019).

Our XAI method is interpretable by design and explains decisions using features and cases. This is similar to the explanation done in case-based reasoning (CBR) (Schoenborn et al., 2021; Gates & Leake, 2021), where decisions on queries are explained by past cases similar to queries. Most CBR systems weight features (Wettschereck et al., 1997), while fewer weight cases (Bicego & Loog, 2016). NN-kNN takes the extra step to train both feature weights and case weights at the same time in an end-to-end manner.

While most modern machine learning methods learn parameters through training, some interpretable AI systems allow parameters to be set using expert knowledge or external methods. For example, in

k-NN, feature or case weights can be determined using mutual information (García-Laencina et al., 2009), the analytic hierarchy process (Bhattacharya et al., 2017), or fuzzy membership functions (Biswas et al., 2018). NN-kNN offers a hybrid approach: feature and case weights are initially learned by the network, then experts can fine-tune them for retraining.

## 3.2 MEMORY AUGMENTED NETWORKS

Weston et al. (2014) proposed the class of memory networks. A memory network consists of a memory $m$ and four components: an input feature map $I$ that extracts features of the query, a generalization process $G$ that updates memory given the query, an output feature map $O$ that produces the output using the query and the memory, and lastly a response $R$ converts output to desired format. They implemented a memory network for question answering purpose. In their example, the query is a textual question, $m$ stores texts, $O$ finds $k$ supporting memories and produces an output feature, and $R$ produces a textual response using RNN on the output feature.

Matching networks (Vinyals et al., 2016) extend the memory networks for one-shot learning. The authors do so by incorporating characteristics from non-parametric models, allowing a trained network to be directly used on a new support set. Similarly, prototypical networks (Snell et al., 2017) learn a metric space and perform classification by computing the distances between the query and prototypes of each class. Li et al. (2018) propose a neural network model that stores auto-encoded embeddings of learned prototypes and makes prediction by comparing the query embedding with the prototype embeddings.

NN-kNN fits the class of memory networks: $m$ is the case layer, $I$ is the feature extraction layer, $G$ is the training of network parameters, and $O$ is the prediction layer based on activated cases. NN-kNN is similar to the matching networks and prototypical networks because the layers until the target activation layer serve as a similarity metric for the cases. In fact, NN-kNN can be considered as a generalization of the previous methods. It works with any case (not just prototypes) or any feature (not just embeddings extracted by an autoencoder). Its k-NN nature allows easy insertion and deletion of cases and easy explanation through activated cases and features.

## 3.3 MENTAL HEALTH DIAGNOSIS BY PRACTITIONERS

The development of efficient diagnosis has stagnated for decades, facing several challenges:

1. **Time-Consuming Diagnostic Processes:** Clinicians continue to rely on diagnostic manuals such as the DSM-V and ICD-10. The time-consuming diagnostic process takes away valuable time from direct therapeutic interventions (Perkins et al., 2018)

2. **Insufficient Training Opportunities:** Only 23% of American Psychological Association (APA)-accredited doctoral programs provide trainee clinicians with the training sites necessary for structured diagnostic interviews (Mihura et al., 2017).

3. **Inadequacy of Diagnostic Manuals:** A systematic review involving 2,228 participants found that clinicians often view diagnostic manuals as unhelpful due to incomplete or inaccurate symptom descriptors (Perkins et al., 2018). They tend to focus on categorizing symptoms rather than identifying the underlying formations of psychological symptoms.

4. **High Rates of Misdiagnosis:** In a study of 309 psychiatric patients, 39.16% patients with severe psychiatric disorders were misdiagnosed, with schizoaffective disorder having the highest misdiagnosis rate (75%), followed by major depressive disorder (54.72%) (Ayano et al., 2021).

5. **Comorbidity and Diagnostic Complexity:** Mental health diagnoses are further complicated by comorbid conditions. For instance, individuals with autism spectrum conditions are more likely to be diagnosed with mental health disorders than non-autistic individuals, increasing the potential for misdiagnosis (Au-Yeung et al., 2019).

Given these ongoing challenges, the field of health service psychology urgently requires a more effective and accurate diagnostic system to support clinicians in making better informed decisions.

### 3.4 Mental Health Diagnosis by AI

Many studies have demonstrated the efficacy of AI-enhanced tools in supporting clinical diagnosis Graham et al. (2019). For example, Zhang et al. (2022) conducted a comprehensive narrative review of 399 studies on NLP applications in mental illness detection over the past decades. Their finding suggested a significant upward trend in research focused on NLP for mental illness detection, with deep learning approaches showing better performance than traditional machine learning methods. In a recent study, Lau et al. (2023) examined deep learning models to automate depression severity assessment by using parameter efficient tuning techniques and pre-trained large language models. Their results suggested that prefix tuning allowed for more efficient model training to reduce over-fitting and offered a scalable solution of automatic depression assessment.

Despite the great promise of AI-assisted clinical diagnosis, both patients and clinicians remain hesitant to accept its application. In one study, patients have expressed a preference for AI tools that are tailored, person-centered, and adaptable to their individual treatment plans and expressed concern that they must adapt to technology (Li et al., 2024). In addition, clinicians often hesitate to fully integrate AI technology into their practice due to concern about their understanding of how AI systems work (Kerz et al., 2023). As such, there is a need for transparency and explainability in AI systems to foster trust with mental health practitioners. However, among the mental health XAI projects surveyed by Joyce et al. (2023), none interviewed specialists about their experience with the XAI models. Surveys in the broader field of XAI (Saeed & Omlin, 2023; Das & Rad, 2020; Molnar et al., 2020) also identified evaluating the explainability of XAI models as a major challenge. In general, there is no agreed upon measure of explainability/interpretability.

In this study, we embrace the idea that explainability is not a rigorous formal concept and that an explanation is only as good as the intended audience perceives it. We take a human-centered approach to evaluate explainability in a qualitative approach that is more common in social science.

## 4 Qualitative Interview and Evaluation

Previous work has already examined the prediction performance of NN-kNN (Ye et al., 2024). This study focuses on the interpretability and adaptability of NN-kNN to aid practitioners in mental disorder diagnosis. We trained NN-kNN on a dataset on the prediction of depression risk and conducted a qualitative study interviewing 10 licensed practitioners about their experience with the model. Meta-parameters for the computational experiment are described in Appendix B.

### 4.1 The Dataset

The model is trained on the dataset from Orozco-del Castillo et al. (2021). The original dataset is answers to a survey of 117 items of true/false answers (see Appendix C). The survey was designed for depression screening according to the Diagnostic and Statistical Manual of Mental Disorders (American Psychiatric Association, 2013) and answered by 157 undergraduate students. After data preprocessing (see Appendix D), our final dataset contains 117 cases; Each case has 102 features of binary values and a class label of 0, 1 or 2, representing low, medium and high risk.

In these experiments, NN-kNN achieves an average accuracy of 0.646, outperforming both standard k-NN (0.417) and LMNN (0.492). We urge caution due to the small dataset size and the unstable results. We did not test extensively on multiple datasets against various models, as previous research (Ye et al., 2024) has already conducted such comparisons. The primary focus of this study is on the interpretability and adjustability of the model for practical use by clinicians.

### 4.2 The Interpretability and Adjustability for Practitioners

NN-kNN is both interpretable and transparent according to the Transparency and Interpretability for Understandability framework (Joyce et al., 2023). Specifically, when predicting depression risk, the model's design emphasizes the following key aspects:

- All cases share the same feature weights ($w_{x\delta_i} = w_{y\delta_i} = w_{\delta_i}$ for any two cases $x$ and $y$), reducing the overall number of parameters. This global feature weighting approach assigns a single weight to each feature across all cases.

- Initial settings for each case $x$ include feature weights set to $w_{x\delta_i} = 1$, case bias at $b_x = 50$, case weight at $w_{(x,L)} = 1$, and class biases at $b_L = 1$. In this configuration, all features and cases start with equal importance. Through training, the model adjusts these weights to improve prediction accuracy. Practitioners can then examine the resulting weights to determine which features or cases the model has prioritized.

- The calculations involved in feature distances, case activations, and class activations rely on basic operations like subtractions and summations. This simplicity ensures that practitioners can easily understand the model's operations. In addition, they can choose to focus on the most or least weighted features/cases for a quick overview of the model's results.

- Each parameter in the model has an interpretable role. The feature weight $w_{x\delta_i}$ and the case weight $w_{(x,L)}$ respectively reflect the relevance of a feature/case to depression. The case bias $b_x$ and the class bias $b_L$ respectively indicate the inherent importance of a case and a class. Practitioners can easily identify outliers or particularly influential features/cases.

- Because each parameter has a specific semantic meaning and a preset initial value, practitioners can manually adjust the feature or case weights based on their expert knowledge. After making adjustments, the model can retrain to further refine the predictions. Practitioners can then assess whether the updated model aligns better with their clinical expertise. We demonstrated this functionality to practitioners and collected their feedback on the adjustable model.

## 4.3 QUALITATIVE INTERVIEW DESIGN

The effectiveness of an explanation ultimately depends on how well it is understood and perceived by its intended audience. To better understand the experiences and perceptions of mental health practitioners, we conducted interviews with 10 licensed clinicians from the US using the interpretative phenomenological analysis (IPA) approach, a widely recognized qualitative research method (Eatough & Smith, 2017). While quantitative methods are prevalent in XAI research—often involving comparisons of models across datasets and using metrics such as prediction accuracy—we chose a qualitative approach for several key reasons:

- **Focusing on a specialized population**:Our target audience consists of licensed clinicians, a small and specialized group. A qualitative approach allows us to zoom in on this specific population, gaining in-depth insights into their experiences with the model, which could be missed in a broader, quantitative study.

- **Tailored model demonstrations**: Demonstrating our model to potential users individually allows us to observe first-hand how each clinician interacts with the system. This personalized approach helps capture the intricacies of their responses, including any challenges they face or specific features they find most valuable.

- **Understanding nuanced reactions**: AI models can evoke a range of responses, particularly when introduced in sensitive fields such as mental health. A qualitative approach is well suited to capturing these nuanced reactions, such as concerns about integrating AI into clinical practice, the potential for AI to aid in diagnostic processes, or hesitations about interpretability. This method allows for a deeper understanding of users' trust and confidence in the model (Maxwell, 2021).

- **Building trust and bridging theory to practice**: By engaging directly with clinicians through interviews, we can address their concerns, answer questions, and foster trust in the model's practical use. This is a crucial step in translating theoretical AI advancements into real-world applications that mental health practitioners are willing to adopt.

Four licensed psychologists at the doctorate level and five licensed clinicians at the master level participated in the interviews, among which four identified as men and six as women. Individually, we demonstrated our model in a Jupyter notebook through Zoom, then invited practitioners to adjust the parameters based on their clinician judgements for depression, followed by 30 minutes devoted for them to answer our qualitative questions. Sample interview questions include: "did you feel like the model has become more useful clinically after you tuned the feature weight?" and "since our model can detect bias, if the model's explanations differ from your clinical judgements, what would you do?" (See Appendix E for a full list of questions).

## 4.4 DATA ANALYSIS THROUGH INTERPRETATIVE PHENOMENOLOGICAL METHOD

Following the four-step analytic process of Interpretative Phenomenological Analysis (IPA), a team of three members (a licensed counseling psychologist, a doctoral candidate in counseling psychology, and an undergraduate psychology student) began by checking each other's biases related to AI and clinical diagnosis. Each member then independently reviewed the interview transcripts, annotating their initial reactions. These annotations were translated into experiential statements, summarizing key aspects of each participant's experience. Next, we compared our individual statements to resolve any discrepancies and clustered them to generate overarching themes. Finally, we compiled eight themes that broadly reflected most participants' experiences. Before finalizing the results, the third author acted as an auditor, reviewing that the themes were grounded in the original transcripts and participants' experiences.

## 5 INTERVIEW FINDINGS

According to our data analysis, eight themes were generated, endorsed by most (at least 6) if not all participants. To remain consistent with the typical way of reporting results for IPA studies (Liu et al., 2020), we combined the themes with our interpretations of participants' experiences to demonstrate a contextual exploration of licensed clinicians' perceptions. To ensure confidentiality, all names used in the following sections are pseudonyms.

### 5.1 THEMES IDENTIFIED AND DISCUSSIONS

**Theme I: Building Trust and Precision in Diagnosis** Participants initially perceived AI-generated diagnoses as inaccurate and unusable, but changed their view after engaging with the adjustable model. After attending our demonstration and adjusting the models themselves, participants reported an appreciation of this model's ability to tune feature weights and adjust cases, allowing clinicians to focus on the most clinically relevant aspects and make diagnoses more precise (Dr. Nate):

> "The tuning process did help somewhat in clarifying the approach to diagnosing by adjusting the feature weights. It allowed for a more targeted focus on clinically relevant aspects."

Participants also emphasized the importance of transparently presenting the model's diagnostic process, which would enhance clinicians' confidence in considering AI-generated results (Dr. Yun):

> "Given that the model is already explainable, and I can see the whole process regarding coming to the diagnosis. It feels more comfortable to understand how it comes to the conclusion. If I want to make some changes to certain features, I can also do that. This gives me more confidence in terms of using it in clinical situations."

Besides, participants observed the adjustment process increased accuracy of depression diagnosis:

> "A general screening can sometimes overlook important clinical features of a client. By being able to adjust the weights on specific features that impact the client's symptoms, I believe the screening becomes more clinically accurate and personalized to the individual, which enhances the overall assessment and treatment planning process."

**Theme II: Potential Risk of Bias Introduced by Model Adjustment** Most participants identified the challenge of maintaining a balance between flexibility and accuracy, expressing concern that their adjustments might lead to less accurate diagnoses (Xing):

> "I am worried about the weight of some questions as I was tuning, which are not the signature of depression...I didn't not quite understand the decision making process of the AI because I would disagree with its clinical decision."

Participants also voiced concerns about the potential risk of introducing bias through adjustments, as clinicians might unintentionally influence the model to confirm pre-existing beliefs (Dr. Yong):

> "The ability to tune the model increases the risk of introducing bias or misusing the tool, especially if there isn't enough transparency about how tuning decisions are made and under what circumstances. Without clear guidelines and a full understanding of the impact of tuning, the potential for bias could compromise trust rather than enhance it."

**Theme III: Customization for Clinical Expertise and Multicultural Factors** Several participants appreciated the ability to adjust feature weights. They emphasized the model's flexibility to accommodate their clinical experiences and theoretical preferences regarding diagnoses (Dr. Nate and Lily):

> "The tunable feature allows for more tailored diagnostic criteria, which can be useful in specific cases, settings, and with particular clients. It offers flexibility, helping clinicians adjust for individual circumstances."

> "It is customized and flexible to fit the patient population (anxiety in kids with or without chronic illness may look different)"

Many clinicians also noted the model's ability to account for multicultural factors, allowing them to adjust for diverse clients and incorporate cultural nuances into the diagnostic process (Dr. Yun):

> "After I tuned the feature weights, the AI model will run again based on my input and generate new features/cases that meets the standards, which is very intelligent. Additionally, regarding the multicultural consideration, clinicians can also tune the features related to clients from diversity background. This is pretty cool."

Furthermore, Dr. Yalin expressed that the model feels like a reliable assistant or skilled trainee, who undergoes personalized training from the clinician to ultimately save time and double-check clinical decisions:

> "I felt like I was training a reliable trainee that, once training completed, can be a great time saver and double check my clinical judgement.

**Theme IV: Transparency and Ethical Trustworthiness** Participants shared their perspectives on the transparency and ethical trustworthiness of the model, which were consistently identified as key factors influencing their willingness to use it in clinical practice. They highlighted the model's enhanced transparency, noting how it distinguishes itself from other AI technologies (Lina and Yaya):

> "It can be ethically reliable because it ensures informed consent from clients and provides transparency in how it influences counselors' decision-making."

> "Additionally, transparency in how AI generates recommendations can further build trust. Ultimately, while trust varies among clinicians, the adaptability of tunable AI can significantly improve its integration into patient care."

However, many clinicians raised ethical concerns about potential misuses, especially if clinicians lack clear clinical or technical guidelines. Similarly, Dr. Yong expressed ongoing concerns about whether the model is truly more ethically trustworthy, despite its technical advancements. She also voiced apprehension about potential confusion regarding the role of AI in replacing human judgment:

> "um...I would say it's better than ChatGPT. I am not sure about 'more ethically trustworthy.' There are still unclear areas seem to be addressed."

> "I think it would be helpful to state that the model is not aiming to take away the human factors when introducing the algorithm. There were moments I got confused from last and this time that the algorithm is trying to take over the human factors in the clinical decision making."

**Theme V: Differential Diagnosis and Time Efficiency** Participants shared other unique strengths of our AI model, especially when it comes to aiding certain differential diagnosis as well as saving time clinically. Before trying out our model, participants had concerns around the depths of AI-generated diagnoses, given that their experiences using generative AI, such as ChatGPT has been frustrating and superficial. Thus, our explainable AI model impressed them with its transparency, resulting in deeper reflections around specific ways of using it for clinical diagnosis.

For example, Dr. Yun shared that in her opinions, AI could assist in quicker and more accurate differential diagnoses, such as distinguishing between Major Depressive Disorder (MDD) and Paranoid Personality Disorder (PPD):

> "Another feature could be helpful is the differential diagnosis, such as MDD/ PPD. If AI can show why it's MDD instead of PDD, that it will certainly help the clinician to spend less time in terms of making diagnosis."

Dr. Yalin disclosed that the traceability and documentation of our model's steps make it a valuable resource for complex decision-making, especially around ethical issues and when dealing with differential diagnoses:

> "because each step can be traced and documented. In our hospital, we often have ethics board meeting to review complex cases (mostly in the medical side) but I can see how having the algorithm can be a supportive data to aid with decision making."

> "Not sure if this is doable - perhaps having a model that can pick up on comorbidity and differential diagnoses? For example, it is sometimes hard to differentiate between childhood anxiety, ADHD, and kids who have both disorders."

Other participants believed that with refinement, our model has the potential to simplify complex tasks and improve diagnostic efficiency, highlighting the excitement around using AI to streamline clinical processes:

> "The potential for simplifying complex tasks and improving diagnostic efficiency is exciting...I look forward to seeing how it develops further and hope that with some improvements, it can greatly assist clinicians."

**Theme VI: Psychotherapy Insights for Future Directions** Along with the excitement of using our AI model for differential diagnosis, some clinicians offered expectations of its future clinical utility. Participants described the ability to track mood symptoms over time offers valuable insights during ongoing therapy, helping clinicians adjust treatment plans based on evolving patient data (Lina):

> "Also, it is beneficial for tracking mood symptoms over time with more thorough contextual insights. I would review the model's explanations and compare them to my clinical insights. Ongoing psychotherapy will allow counselors to gain a deeper understanding of their clients, enabling them to enhance their clinical judgment and make more informed decisions moving forward."

Other participants thought that AI could enhance diagnostic reasoning by flagging symptoms that don't fit a particular diagnosis. Dr. Yun expressed that AI can not only show why it supports a diagnosis but also highlight why it rules out others like anxiety or obsessive-compulsive disorder:

> "I was thinking if the AI can also add some features, such as flagging some symptoms that's not aligning with diagnosis (i.e. feeling very energetic in everyday life - which is not typical for a depression diagnosis), this could also help the clinician see why the AI make this diagnosis. It's like not only showing the key symptoms that make AI decide that this case fits diagnosis, but also symptoms that makes AI decide it's not anxiety/OCD/eating disorder, etc. This could also be helpful for the clinicians to see why AI rule out these other diagnosis."

Last but not least, one participant (Xing) suggest the model to capture key mental health factors, like affect and history, and summarize the client's mental health background:

> "I think there are so many important factors that the model is not detecting yet. eg. affects, history of mental health. I would also be curious of a feature that could summarize client's mental health history."

**Theme VII: Peer Consultation and Training for Clinicians** Two participants discussed the potential utility of the model for peer consultation and clinical training. Clinicians began to envision how they might incorporate the model into their decision-making processes, suggesting a growing willingness to see themselves as potential users.

Clinician Xing mentioned that if the model's output differed from their judgment, they would seek guidance from peers or supervisors, acknowledging the importance of human expertise in complex clinical cases:

> "I am worried about the weight of some questions which are not the signature of depression, but it also adds in personal bias to the tool. I would seek peers/supervisors for a second opinion when the model differs from my clinical judgement."

Clinician Lina discussed that effective training on model tuning is crucial to avoid bias and ensure the model's appropriate use. Ongoing guidance would help clinicians incorporate evidence-based practices into their work:

"Will there be more guidance on using this tool to enhance clinical judgment or incorporate evidence-based criteria? Counselors' biases can be identified by comparing similar clients' depression symptoms and analyzing their similarities and differences."

**Theme VIII: Varied Potentials in the Future** When considering the future of the model, participants expressed a mix of concerns and excitement. Two participants discussed the importance of having an user-friendly and visual interface to enhance the user experience, allowing clinicians to see the impact of weight changes clearly. This would improve engagement and the overall utility of the model:

"I would like to make this model more user friendly, visual, and show more clear changes/interpretation after each feature weights change."

"Ensuring that the tool remains user-friendly and offers high fidelity in its outputs is crucial for its future success. I look forward to seeing how it develops further and hope that with some improvements, it can greatly assist clinicians."

Clinician Wang raised concerns about allowing all users to adjust the feature weights, questioning whether this flexibility might compromise accuracy. Other participants echoed this concern, worrying that over-manipulation by clinicians could cause important peripheral data to be overlooked, potentially leading to inaccurate diagnoses:

"I am not sure what's the usage of this model. It's like allowing every clinician to change the algorithm, than how is that applicable to large amount of clients?"

"Over-manipulation by the clinician that may overlook the importance of some peripheral data. It's adjustable but not sure how accurate the result is."

We observed differences between doctorate- and master-level clinicians during the interviews. Doctorate-level clinicians were more proactive, asking clarifying questions, adjusting features creatively, and exploring how to integrate the model into their work. In contrast, master-level clinicians were more defensive, raising concerns that led to shorter, less in-depth interviews. These differences may be due to doctorate clinicians' broader roles and theoretical training, while master-level clinicians, burdened by heavy caseloads and limited support, focused more on practical applications and found it difficult to engage with the prototype without a polished user interface.

## 6 Conclusion

We propose NN-kNN for mental disorder diagnosis not as a solution to the inherent challenges of diagnosis, nor solely for its predictive accuracy. All models, including NN-kNN, are biased because they are at most as effective as the data they are trained on. What sets NN-kNN apart is its full interpretability and adjustability, enabling practitioners to detect and correct biases within both the model and the data. By offering the ability to adjust model parameters based on expert knowledge, NN-kNN empowers clinicians to make more informed and ethical decisions. Practitioners can manually adjust the model parameters, discover unforeseen patterns or biases, and decide when it is appropriate to rely on AI in clinical settings.

Through our qualitative interviews, practitioners expressed appreciation for the model's transparency, flexibility, and the way it integrates their clinical expertise into the diagnostic process. This balance of AI-driven insights and human judgment has the potential to build greater trust and utility in AI-based diagnostic tools. Looking ahead, we plan to extend NN-kNN by incorporating complex data from additional modalities, further supporting clinicians in delivering more holistic and data-rich diagnoses.

AUTHOR CONTRIBUTIONS

Hidden for Anonymity

ACKNOWLEDGMENTS

Hidden for Anonymity

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

## A    Appendix

You may include other additional sections here.

Authors may use as many pages of appendices (after the bibliography) as they wish, but reviewers are not required to read the appendix.

## B    Meta-parameter Settings

We performed a standard 10-fold cross-validation on the dataset, training the model until test set accuracy plateaued for a fixed number of epochs (40). Due to the small dataset size and the fact that the goal of this experiment is not to precisely gauge the model's prediction accuracy, we did not allocate a separate validation set. The model was trained in batches of 4 using the Adam optimizer with a learning rate of 0.01. The top-$k$ case selection layer was disabled, as enabling it led to lower accuracy. Although some parameter choices (e.g., the relatively high learning rate) may seem unconventional in typical machine learning setups, they are justified and discussed in Ye et al. (2024). Most importantly, additional configurations were employed to enhance the model's interpretability for practitioners:

- All cases share the same feature weights ($w_{x\delta_i} = w_{y\delta_i} = w_{\delta_i}$ for any two cases $x$ and $y$) to reduce the number of parameters.This approach applies global feature weighting, where each feature is assigned a single weight across all cases.

- For each case $x$, we initialize the feature weights as $w_{x\delta_i} = 1$, the case bias as $b_x = 50$, the case weight as $w_{(x,L)} = 1$, and all class biases as $b_L = 1$.

## C    Dataset Description

The dataset is a collection of answers from 157 undergraduate students to a survey of true/false questions. The questions are designed according to the Diagnostic and Statistical Manual of Mental Disorders by a psychologist (Orozco-del Castillo et al., 2021). The questions are listed below.

1. Most of the time I have difficulty concentrating on simple tasks

2. I don't feel like doing my daily duties

3. My friends or family have told me that I look different

4. When I think about the future it is difficult for me to imagine it clearly

5. People around me often ask me how I feel

6. I consider that my life is full of good things

7. My hobbies are still important to me

8. I'm still as punctual as I have always been

9. If I had the chance, I would spend all day in my bed

10. I have found that I can spend a lot of time scrolling the screen of my cell phone without searching or stopping at anything in particular

11. When someone asks me something, I have noticed that I take longer than normal to respond

12. I have noticed my body shaken without any cause

13. I felt more encouraged to do my daily activities before

14. Sometimes I wake up sad and I can't explain why

15. In recent months I usually reproach myself for things from the past

16. I think my thoughts are strange or different from before

17. I feel guilty about the decisions that I have made

18. I don't feel as comfortable with my body as I did before

19. I don't feel successful compared to others

20. It is difficult for me to make decisions even if they are simple

21. I'm capable of achieving what I propose to myself

22. It is not difficult for me to understand something the first time

23. I have thought more than before about what my death would be like

24. Being dead seems to be a solution to some problems

25. I would rather stay home than go out with my friends

26. I like to attend family gatherings

27. I feel excited when thinking about my life project

28. The decisions I have made so far have been the right ones

29. I am able to carry out my activities as I have always been

30. I like to be in touch with my friends and family through social media

31. It is easy for me to choose a photograph of myself to show it on social media

32. I am proud of what I have achieved so far

33. I have trouble remembering things easily

34. In recent months I have had discussions with my schoolmates or colleagues

35. I constantly imagine that something will go wrong at my work or at school

36. I am afraid of being wrong when doing my homework

37. I'm not too worried about what might happen in a few weeks

38. Lately it's hard for me to calm down

39. Everything will be alright

40. I can easily blank my mind

41. I am bothered by insignificant things that were not important before

42. I find it uncomfortable to be in a crowded place

43. Sometimes I feel trapped

44. I am easily frightened by unexpected noises

45. I have difficulties to do one task at a time

46. I have the feeling that I am forgetting to do something

47. I can clearly express to others how I feel

48. I can sleep easily

49. I enjoy every moment of the day

50. I imagine that at any moment a disaster of nature may occur

51. Sometimes I feel like I get tired easily

52. Being locked in an elevator would be the worst thing that could happen to me

53. I'm bothered by people walking slowly in front of me

54. I don't usually get upset if something doesn't go as expected

55. Sometimes it is as if some conversations with friends or family become interrogations

56. I manage my schedule as I always have

57. It bothers me to feel that people on the street approach me

58. I have no difficulty understanding what people explain to me

59. I consider that I am good at controlling my emotions

60. In new situations I feel calm and encouraged

61. Sometimes I forget what I wanted to say because I have several thoughts at the same time

62. I would like to know what will happen in the future

63. When I get angry I can easily explode

64. I can put down my cell phone and dedicate myself to reading without distractions

65. I worry that people will not understand what I mean

66. Sometimes I do not listen to what people say to me because I am thinking about other things

67. I get angry easily

68. I'm afraid that something bad could happen to me

69. It is not important for me to meet set dates

70. I like to think clearly before giving my opinion

71. I use lies just to get out of certain problems

72. If I have the opportunity to get in line to avoid wasting time, I do it

73. I have difficulty making elaborate plans

74. People have problems because of themselves

75. No me parece importante lo que los otros piensen sobre mí

76. Laws are not as important as others think

77. I would regret betraying a friend

78. I prefer that a negotiation supports the largest possible number of people involved

79. It is easy for me to work in a team

80. It is important to help people when they need it

81. I have punched someone or thought of doing it

82. If it was necessary I would pretend to be someone else to get something

83. I consider it important to ensure my physical safety and that of those around me

84. After an argument I usually go over what happened in my head

85. I have a hard time controlling myself when I get angry

86. Loyalty is important

87. If I can help a person I will stop what I'm doing to help them

88. Sometimes people need physical force to understand

89. It makes me laugh when my superiors at school or at work demand something

90. Deceiving people is not wrong if it is to achieve something important

91. I like to greet my neighbors

92. It does not seem serious to me to have some debts

93. People steal because they have needs

94. I lose control easily

95. Neighbors must put up with each other's noises without complaining

96. Littering on public roads is wrong

97. People who commit crimes have their reasons for doing it

98. It is normal to change jobs several times a year

99. It is important to respect turns

100. I could pretend to be someone else to achieve what I want

101. I consider it important that all people have the same rights

102. I have a hard time taking "no" for an answer

## D    DATA PREPROCESSING

The dataset was used to train neural networks for depression screening and then later used for an explainable AI challenge in the Explainable AI Challenge at the 2022 International Conference on Case-Based Reasoning (Wilkerson et al., 2022). We obtained a version of the dataset with 104 cases with 102 attributes and a risk score (from 1 to 5) calculated from the number of physical symptoms related to depression. We oversampled less frequent classes to counteract class imbalance and transformed risk scores into three classes (low, medium, and high risk of depression) following the example of Wilkerson et al. (2022). Our final dataset contains 117 cases; Each case has 102 features of binary values and a class label of 0, 1 or 2, representing low, medium and high risk.

## E    INTERVIEW QUESTIONS

Following are the questions we used when interviewing the 10 participating practitioners.

### E.1    DEMOGRAPHIC QUESTIONNAIRE

1. What's your full name?

2. What's your gender?

3. What's your age?

4. What's your race?

5. What's your sexual orientation?

6. What's your highest degree and for how long have you been licensed?

7. What's your professional field (Counseling, clinical, social work, school psychology, etc.)?

8. What's your current job?

9. Which state are you currently living in?

10. What are your primary clinical populations?

11. What are your primary theoretical orientations?

12. Please rate your knowledge about AI technology from 0-10 (0 being absolutely no knowledge, 10 being complete expertise).

### E.2    QUALTRICS QUALITATIVE QUESTIONS

**As the preliminary user:**

1. Regarding the tunable experience you just had, did you feel like the model has become more useful clinically after you tuned the feature weights? Please elaborate.

2. Did you feel like the model has become more clinically trustworthy? Please elaborate.

3. Please provide any other comments or thoughts about your experience trying out our algorithm just now.

**As a practitioner thinking about using this model for clinical diagnosis:**

1. If you were to use this AI algorithm, would the tunable feature be useful for you?

2. What would be the strengths and concerns you have about using this tunable feature?

3. Since our model can detect bias, if the model's explanations differ from your clinical judgment, what would you do?

4. Since our model is fully interpretable and tunable, would this algorithm be more ethically trustworthy?

5. What other factors should we consider in improving this model for clinical diagnosis?

