# OpenReview forum: "All Models are Biased, Some are More Transparent about it: Fully Interpretable and Adjustable Model for Mental Disorder Diagnosis"
_ICLR.cc/2025/Conference — Submitted to ICLR 2025_

### Official Review · Reviewer_SJtk · 2024-10-28

**Soundness:** 2
**Presentation:** 3
**Contribution:** 2
**Rating:** 5
**Confidence:** 4

**Summary:**

The paper introduces the reader to the problem of black box models for mental-health issues, which medical professionals cannot use well because they do not offer the explainability of their decisions, with post-hoc methods not being able to fully explain how the decision of a model was made. To solve this they propose the use of a recent method NN-kNN. They use the model to qualitatively show how the experience of the medical practitioners’ changes by using this model and offer insight into the clinicians’ experience.

**Strengths:**

- Real-world applications of XAI are a very important and current topic.
- I really appreciate the evaluation on users that are a decent representation of the end-user of such an approach! This is severely lacking in XAI at this moment. Even with the compromises that have to be made

**Weaknesses:**

1. Since the NN-kNN model has already been quantitatively assessed by related work, the main contribution of the paper is the qualitative study.

2. The interpretation of the study’s results does not align with the results. The model’s strengths, specifically how each parameter of the model can help the practitioners in explaining the relevance of features to depression, seem to justify the use of the model in general. The general opinion is that the model can be useful in some specific cases however they would not generally trust it, specifically because while it is explainable, changing the parameters seems to help sometimes and is counterproductive other times.

3. It seems like changing the model’s weights to align more with clinicians’ beliefs confirms the clinicians’ bias even more (this issue is also raised by Dr. Yong). While your study provides novel results, the conclusions you draw are showing support of the model even though the corresponding clinician’s opinions mostly do not confirm that. You should be more critical of the model’s ability in helping a clinician. It seems like you agree with this in the first paragraph of the conclusion, however this is not visible from section 5 and from the last paragraph of the conclusion, where you again show strong support for the use of the model.

4. We have to at least acknowledge two limitations of the study methodology in the context XAI.
- There is more and more evidence that participants perception of how a XAI method helps is not really that correlated with actual performance. A good starting point: Amarasinghe et al.: "On the importance of application-grounded experimental design..."; Bucinca et al: "Proxy tasks and subjective measures...".
- Without a placebo or baseline method, we can't really know if the proposed method performs well because of the mere addition of some XAI method or if it performs better than any of the methods criticised in the paper. A good starting point: Eiband et al.. The impact of placebic explanations on trust in intelligent systems.

Therefore, a placebo (or at least some baseline) quantitative study with some real-world performance metric would be the next step.

Minor comments:
- A period is missing at the end of the sentence on line 197.
- The sentence in line 251 “The original dataset is …” could be rewritten.
- Line 299: Space is missing after “:”.
- In line 422, and the whole section, the use “our” model is misleading, because the model was not created by the authors but was only evaluated.

**Questions:**

- The details of the study protocol are vaguely described in different parts of the paper or missing (How were they recruited? Demographics or any other meta-data?). I would prefer if the study was described in one place and in enough detail so it is reasonable to claim that it can be reproduced. Even if it all ends up in the Appendix.

---

> ### Author Response · Authors · 2024-11-14
>
> Thank you for your detailed review! You are the only reviewer who actually commented about the qualitative study component. We really appreciate that. Other reviews we received have shown that there is a misunderstanding of why a qualitative study matters for XAI in the first place.
>
> Please allow me to address some of the weaknesses and question here.
>
> "The interpretation of the study’s results does not align with the results. " "It seems like changing the model’s weights to align more with clinicians’ beliefs confirms the clinicians’ bias even more (this issue is also raised by Dr. Yong)."
>
> There is a misunderstanding here. The practitioners have mixed review about the model's results, about this specific setting of the model on this specific data set, but they are able to make such judgements because of the interpretability of the model. In other words, a practitioner may appreciate the model's transparency and flexbility, and then decided to not use it because they think the model is biased and or can be biased by their own tuning for a certain domain task. The first paragraph of conclusion discusses this while the last paragraph talks about more positive things and potential future direction.
>
> "There is more and more evidence that participants perception of how a XAI method helps is not really that correlated with actual performance. "
>
> Thank you for your lead on the related work here! It is indeed our future goal to build real applications in clinic setting. Because of the sensitivity of the task, it deserves to be a standalone project. And mental health diagnosis differs a lot by location, culture, the disorder we are targeting, ethics, law and etc., so one application built for one clinic site for one disorder may not be useful for other contexts. Therefore the current study is a *necessary* open-ended exploration before the application step.
>
> "Without a placebo or baseline method, we can't really know if the proposed method performs well because of the mere addition of some XAI method or if it performs better than any of the methods criticised in the paper"
>
> This is partially limited and also partially intended by the qualitative research design  (We have another study where we have users compare and disucss about the results from black-box models and the white-box models). So our goal here is not show that we are *better* than some other methods, but to show psychologists' reactions and thoughts on one example of interpretable AI. We think that comparing our model with another model will unavoidably and unfairly cast favor on our model and interfere the interviewee's perspectives. For example, if we ask the question "this is a black-box model A, this is a somewhat white-box model B, and this is our fully white-box model C, which would you like", or "rank them in Likert scale", I think the answers collected will offer less insight into the XAI usage, than our current design. This actually relates to our last point: "There is more and more evidence that participants perception of how a XAI method helps is not really that correlated with actual performance. " A quantitative study may easily be an ivory tower that is not related to how mental disorder diagnosis is actually carried out.
>
> Therefore we decided a full qualitative research design for this study.
>
> “The details of the study protocol are vaguely described”
>
> Thank you for this suggestion, we will add an appendix section discussing all details together.

---

### Official Review · Reviewer_uJfq · 2024-11-04

**Soundness:** 3
**Presentation:** 3
**Contribution:** 1
**Rating:** 3
**Confidence:** 3

**Summary:**

The paper proposes to use a neural network based k-nearest-neighbor algorithm (NN-kNN), i.e., a k-nearest neighbor algorithm, that can explain each model decision with activated cases, and each activated case can be attributed to its feature distances with the query.
As said by the authors "This study focuses on the interpretability and adaptability of NN-kNN to aid practitioners in mental disorder diagnosis."

Thus the main issue with the paper is that, even if very interesting as results, it only consists on the usage of NN-k-NN on a case study with interviews with prectitioners related to the interpretability. As a consequence, I do belive that ICLR is not the right venue for this contribution as it lacks of methodological novelty.

**Strengths:**

interesting as results

**Weaknesses:**

it only consists on the usage of NN-k-NN on a case study with interviews with prectitioners related to the interpretability

**Questions:**

Any novel contribution with the exception of the specific application?

---

> ### Author Response · Authors · 2024-11-14
> **About methodological novelty**
>
> Thanks for the review!
>
> Our paper fits in the category of interpretable and explainable AI. Based on the [CfP](https://www.iclr.cc/Conferences/2025/CallForPapers): "We consider a broad range of subject areas including feature learning, metric learning, compositional modeling, structured prediction, reinforcement learning, uncertainty quantification and issues regarding large-scale learning and non-convex optimization, as well as applications in vision, audio, speech, language, music, robotics, games, healthcare, biology, sustainability, economics, ethical considerations in ML,"
>
> Our work will fall in the category of healthcare and ethical considerations in ML.
>
> In the "non-exhaustive list of relevant topics": our work is relevant as "metric learning", "interpretation of learned representations", "societal considerations including fairness, safety, privacy", and "neurosymbolic & hybrid AI systems" (because NN-kNN is a hybrid of neural network and case-based reasoning)
>
> We understand the feeling that this does not look like a ML paper, because we do not evaluate it using traditional quantitative ML approach. Instead we evaluate this model using a qualitative approach that is widely accepted amoung counseling psychologists, because we are discussing about model transparency and trust for psychologists as future users.
>
> So we think that our work is indeed a novel model with intellectual merit and also contribute to the XAI community by show casing a qualitative study combining AI and mental disorder diagnosis.

---

### Official Review · Reviewer_apt5 · 2024-11-07

**Soundness:** 2
**Presentation:** 1
**Contribution:** 4
**Rating:** 1
**Confidence:** 5

**Summary:**

The paper discusses the problem of interpretability and explainability with AI, and studies the use of a neural network-based k-nearest-neighbor algorithm (NN-kNN) for diagnosis of mental illnesses and for interpretability and explainability through conducting a qualitative study.

**Strengths:**

The paper does address an important area of interest in psychiatry and AI. XAI is something that interests both mental health professionals and AI experts. With the paper studying NN-kNN, it does further contribute to the discussion of XAI and use of deep learning and machine learning in psychology and psychiatry. It also illustrates the use of a deep learning/machine learning model (NN-kNN) in diagnosing mental disorders, which is a very good contribution as it shows the use of a newly created AI model in psychiatry.

**Weaknesses:**

There are many significant issues with both the presentation and the soundness of the paper. There is also a concern tied to the soundness, which can impact the contribution of the paper.

In terms of presentation, the paper was confusing to follow. This is due to the way that the paper is organized and the writing. I will mention only a handful of issues that contributed to the confusion. The paper would have been better organized if the authors swapped Section 2 and Section 3. That is, talk about the related work first, then talk about the model that the authors used. Additionally, it is very unclear why the authors went with NN-kNN instead of other interpretable machine learning algorithms. Also, not only does the title of the paper not make sense, but it doesn’t connect to the paper at all. More details on this will be provided in the next paragraph on soundness. Another thing is that since NN-kNN is considered both a NN and a kNN, the authors should include literature on the use and studies of kNN for mental illness diagnoses in the related work section. Doing so would help with understanding what work has been done for kNN and why kNN model would be suitable for mental illness diagnosis. One other thing is that the paper could use another round of proof reading. There are sentences that are written incorrectly. For example, it’s supposed to be “then invited practitioners to adjust the parameters based on their clinical judgements for depression” and not “then invited practitioners to adjust the parameters based on their clinician judgements for depression”. Lastly, in the introduction of the paper it says that they will conclude with discussions and future directions, but in the conclusion only one point is made about future directions and the things mentioned in the discussion are confusing. For example, in the conclusion it states, “We propose NN-kNN for mental disorder diagnosis not as a solution to the inherent challenges of diagnosis, nor solely for its predictive accuracy”. This is very confusing as the sections before that says otherwise. From what I understand, the sections before say that NN-kNN is a solution for mental disorder diagnosis because of its adjustability, interpretability, and explainability. I think the authors meant “We propose NN-kNN for mental disorder diagnosis as a solution to the inherent challenges of diagnosis, nor solely for its predictive accuracy”, but I can be wrong.

In terms of soundness, the paper has several serious issues that need to be addressed before it can be published anywhere. I can’t mention all of them as that would make this review very, very long. Instead, I will just mention and detail two severe issues. One severe issue is the paper claims that “all models, including NN-kNN, are biased because they are at most as effective as the data they are trained on”. While this is true, the paper doesn’t adequately support the claim with any sort of evidence or anything. The paper only covers one model, and even for that the focus wasn’t on the biasness associated with the model, it was on its interpretability and explainability. In addition, there isn’t any discussion or anything about models being bias in the paper. It’s just mentioned once in the title and then once in the conclusion, which is a major problem. Not only does it make the paper confusing, but it also impacts the objective and cogency of the paper. This claim also raises a concern, which will be mentioned in the final part of this review. Because of the lack of coverage and studying conducted on the biasness of models, the title doesn’t make any sense. I would suggest removing “ALL MODELS ARE BIASED, SOME ARE MORE TRANSPARENT ABOUT IT” from the title and the parts of the paper that mentions biasness of models.

Another severe issue with the paper comes from the claim the authors make in introduction, which is that the paper “introduces a novel approach to human-machine interaction drawing on insights and methodology from both AI and psychology”. This novel approach from what I understand is supposed to be the XAI approach. The paper itself doesn’t discuss the XAI approach at all. Instead, the paper presents an XAI model, shows its application in mental health, and shows its interpretability and explainability through a qualitative study. XAI approaches and XAI models are two different things. To help understand what I mean from this, I’ll use an existing XAI study. In the paper “Explainable AI meets Healthcare: A Study on Heart Disease Dataset” (URL to article: https://arxiv.org/abs/2011.03195), the authors do something similar to what the authors of the paper being reviewed are trying to do but for a AI model focusing on heart disease instead of mental illnesses. In that paper, they clearly state the types of AI models (XGBoost), and XAI approaches (contrastive explanation methods, example-based techniques, etc.) they will use in their study, and then discuss them accordingly. Going back to the paper being reviewed – the paper states the AI model used, but doesn’t state, explain or detail the XAI approach used. Consequently, the paper does not introduce a novel approach to human-machine interaction like the paper claims it does. Based on the qualitative study findings, it could be that the XAI approach that the authors are talking about is: 1) present the model to the clinicians; 2) let the clinicians adjust the parameters based on their clinical judgements for the mental disorder; 3) conduct a qualitative study with using the interpretative phenomenological analysis (IPA) approach. If that is the case, then it goes back to the presentation issues, which is that the way that the paper is written is in a way that does not make that clear. To make this clear, not only does the wording and structure of the paper need to be changed, but the paper should also include a part/section that covers the name of the XAI approach, an explanation, and enough details for others to be able to clearly understand and use the approach.

The final part of this review is on a noteworthy concern with the paper. The concern is that if the paper is on how all models are biased and that some are more transparent about it than others like the title of the paper and the conclusion say it is, then that would make contribution of the paper worthless. It is already known within the AI, computer science, engineering, statistics, etc. communities that all AI models are biased because they are at most as effective as the data they are trained on, and that all AI models are not equally transparent about it. To prevent the paper from losing its value, I would recommend to the authors to not focus on that at all and instead focus on either NN-kNN being used as an AI algorithm suitable for mental illness diagnosis or the approach to human-machine interaction/XAI approach that they were trying to present.

**Questions:**

What's the exact objective of the paper? Is it to introduce a new XAI approach? To introduce a new AI model that explainable and can be used for mental disorder diagnosis? Or to show that all models are biased and that some models are more transparent about it than others?

Based on the information you provided, do you think that your study is easily reproducible? That is, do you think others will be able to easily conduct the experiment and get the same or similar results as your study did?

For suggestions on how to improve the paper, please see the weakness section.

**Details Of Ethics Concerns:**

Not applicable.

---

> ### Author Response · Authors · 2024-11-14
>
> Thank you for the detailed review! You pointed out a few issues that we think we can address with editing. We wish to discuss about some of the other comments, I put yours in quote so it's easier to read.
>
> As for the questions:
>
> 1. The goal is to carry out a tailored model demo for a highly specialized group, and understand their need to build trust in XAI.
>
> 2. The model implementation and experiment result can be fully reproduced (will share on github after publication). The IPA qualitative approach is weakly reproducible by [nature](https://cheresearch.org/interpretative-phenomenological-analysis/interpretative-phenomenological-analysis/), because if you interview another person, they will say something different. But the interview results are nevertheless valuable, as discussed in section 4.3 QUALITATIVE INTERVIEW DESIGN
>
> **About presentation**
>
> We originally had related work on section 2 but changed it to the current flow, because:
>
> Section 2 explains NN-kNN -> Section 3 explains related work in AI -> Section 3 also explains related work in mental health -> Section 4 continues about evaluating on mental health diagnosis -> And the rest of the paper focusing on psychologists' discussion. So the whole paper flows from AI point of view to psychology point of view. We can, of course, revert back to a more traditional flow with related work goes before everything else.
>
> "the authors should include literature on the use and studies of kNN for mental illness diagnoses in the related work section"
>
> We include some related survey in section 2.1 EXPLAINABLE AND INTERPRETABLE AI (section number might be different now since we are reordering). And there are very few kNN for mental illness diagnoses, understandably so because other methods often outperform kNN and probably is more sophisticated. For example, [this survey we cited](https://www.nature.com/articles/s41746-023-00751-9#:~:text=The%20literature%20on%20artificial%20%EE%80%80intelligence%EE%80%81) listed only one study using kNN. This relates to the next point:
>
> "Doing so would help with understanding what work has been done for kNN and why kNN model would be suitable for mental illness diagnosis."
>
> This is discussed in section 4.2 THE INTERPRETABILITY AND ADJUSTABILITY FOR PRACTITIONERS
>
> You also raised a few very valid edits. We really appreciate that and made those corrections. For example, the first sentence in conclusion now reads "We propose NN-kNN for mental disorder diagnosis not as a solution to the inherent challenges of diagnosis, but as a tool to enhance understanding and transparency in the diagnostic process."
>
> **About soundness**
>
> You mentioned: "One severe issue is the paper claims that “all models, including NN-kNN, are biased because they are at most as effective as the data they are trained on”. While this is true, the paper doesn’t adequately support the claim with any sort of evidence or anything."
>
> You also mentioned later: "It is already known within the AI, computer science, engineering, statistics, etc. communities that all AI models are biased because they are at most as effective as the data they are trained on." This is exactly why we didn't expand on "all models are biased" part.
>
> "The paper only covers one model, and even for that the focus wasn’t on the biasness associated with the model, it was on its interpretability and explainability. In addition, there isn’t any discussion or anything about models being bias in the paper."
>
> I can understand why this is an issue. In our experiment, we did notice a few biases learned by the model. We did not list such details on biases, because it's too specific for the data set and cherry-picking. Instead, we list the high-level themes based on interview transcripts, because the bigger picture is more important for future research on this. And there are some discussion about bias, for example in Theme II and Theme VII.
>
> "I would suggest removing “ALL MODELS ARE BIASED, SOME ARE MORE TRANSPARENT ABOUT IT” from the title and the parts of the paper that mentions biasness of models"
>
> With that being said, I can understand your viewpoint. Our title is based on a [quote](https://jamesclear.com/all-models-are-wrong), only to intrigue reader. We do not intend to show "all models are biased", as no one can.
>
> **About terminology of XAI approach and XAI model**
>
> You raise a good point about XAI approach and XAI model. What we intend to do really is the XAI model in your definition. We showcase NN-kNN as an XAI model in mental health diagnosis. We acknoledge that we were not aware of this terminology difference and will make changes accordingly. I am transitioning from my field to XAI recently, and I have been using approach and model semi-interchangeably.
>
> We hope you can reconsider the rating based on this discussion. We also welcome further feedback and discussions.
>
> Thanks again for detailed review!

---

> > ### Comment · Reviewer_apt5 · 2024-11-27
> >
> > Thank you for the detailed response. I have changed the rating for the soundness from poor to fair. The only reason the rating didn't get higher than that is because of the ongoing presentation problem. If you reword and change claims and other things made in the paper then the rating can change to a better score.
> >
> > Reading your comment has made me realize that a lot of the problems stems from the way the paper is presented. If the paper can be changed to word things better, then the paper can potentially get accepted. However, the way it is now makes it such that most reviewers would reject it, and most readers would not benefit from it as much as they could. Also, just know that there are in fact a lot of studies on kNN for mental health diagnoses. I did a very quick search on Google Scholar and easily found 3 papers that study kNN use for mental disorder diagnosis. This first paper studies several ML algorithms for mental disorder diagnosis, one of those algorithms being kNN: https://iopscience.iop.org/article/10.1088/1742-6596/2161/1/012021/meta. This second paper studies also does the same thing, but it was published this year: https://www.scienceopen.com/hosted-document?doi=10.57197/JDR-2024-0022. Lastly, this paper reviews of ML algorithms for diagnosing mental illnesses using existing studies: https://pmc.ncbi.nlm.nih.gov/articles/PMC6504772/. Note that in the paper they highlight kNN being one of the ML algorithms that is frequently used for diagnoses and other areas of mental health and psychiatry. There are many more literature out there that say the exact same thing, and there are enough published studies out there to support it. It is true that other methods often outperform kNN, but kNN has been and still is studied and used for showing ML's diagnosis of mental disorders and comparing it's performance with other ML algorithms. I would encourage the authors to read more about ML in psychiatry to get a better understanding of AI and ML in psychiatry, and get a better sense of how to address issues with the presentation of the paper.
> >
> > It's best that the authors work on the paper and the either re-submit the paper to ICLR next year or to another venue.

---

### Official Review · Reviewer_1eMZ · 2024-11-07

**Soundness:** 2
**Presentation:** 2
**Contribution:** 2
**Rating:** 3
**Confidence:** 3

**Summary:**

In this work, the authors study the interpretability and adaptability of neural network-based model NN-KNN. Because each parameter in the model has some semantic meaning, and the network relies on simple operations, it is interpretable. They then conduct a qualitative study about the utility of this model in mental disorder diagnosis.

**Strengths:**

- Paper is well-written and easy to follow
- The authors carry out a detailed qualitative study about how clinicians and experts could utilize such a model for diagnosis of a mental disorder

**Weaknesses:**

- The sample size for training the model seems relatively small (117 cases), which might make the model less reliable. Additionally, the performance of the model is low. It does seem like there are larger datasets with similar questionnaire-based case-level answers and scores (for example, the DAIC WOZ dataset with PHQ scores/sub-component scores)
- It is not clear how the much the simplifying assumption of a global feature weighting (which makes the model interpretable) impacts performance
- The questions in the questionnaire seems particularly focused on the tunable weights component of the model, not the broader interpretability and trust
- The study might have been stronger if the authors compared with another potentially non-interpretable but simple model (like a decision tree or logistic regression model, all of which adjust their weights/structure during training)

**Questions:**

- Can authors clarify the impact of the global feature weighting assumption on performance?
- Can authors justify the use of a relatively small dataset in their study?

**Details Of Ethics Concerns:**

Can authors clarify if IRB approval for the study was obtained?

---

> ### Author Response · Authors · 2024-11-13
> **Clarification on the feature weighting and data sets.**
>
> Thank you for the detailed review!
> Your comments regarding the weaknesses and questions are very relevant. We discussed some in the paper but we are limited by space, so please allow me to elaborate a bit here.
>
> About the impact of the global feature weighting:
>
> Thank you for raising this question. It is a very relevant techincal aspect of the model. With feature weight sharing disabled, the model achieves an accuracy of 0.656 (very similar to when feature weight sharing is enabled). We decided to not discuss the setting of feature weight sharing because keeping it on will have better interpretability. And the paper is less about an accuracy number, more about interpretability.
>
> About the data set:
>
> Unfortunately that was the most suitable data set we can find that is publicly available. You suggested the DAIC WOZ dataset with PHQ scores/sub-component scores. We did look into that one, but it is a audio/video/transcription data set of interviews. It contains eye tracking, facial tracking, and textual transcription, which are currently not the goal of our project. It also contains the PHQ questionaire data, but it has other issues. It contains about 200 samples. Each sample has 8 questions that is directly related to depressive symptoms, a score that is the sum of the 8 questions, and a final label whether the score is more than 10. So it is highly entangled (the label is just a sum of the features) and is actually trivial compared to the data set we use.
> We also looked into the data set in related literature but they are almost all behind some restriction and not publicly available.
> We thought about reaching out to hospitals/counseling clinics but we have yet to reach any institution willing to share such sensitive information. It is in our intention as a future direction to get a better data set and potentially collaborate with real clinic.
>
> About the model's accuracy:
>
> on a related note, a neural network classifier with 3 hidden layers achieve an accuracy of 0.581 on this dataset (10-fold crossvalidation average). Part of the reason why we choose this data set (over a couple others we can find where we can achieve high accuracy) is that it does reflect real diagnosis problems as discussed in the paper (Diagnosis by human has an accuracy of 54.72%). The data set poses many interesting questions that actually turn out to be useful in predicting depression even to some psychologists' surprise.
>
> About broader interpretability and trust:
>
> Indeed, our model is interpretable for its transparency and its tunable weights, but I would say half of the eight questions we asked are still about the broader interpretability and trust. This is also reflected in the themes in the interview findings, where at least half of the themes are about interpretability and trust in general.
>
> "The study might have been stronger if the authors compared with another potentially non-interpretable but simple model":
>
> We are limited by the qualitative design here. Where we use majority of our effort in interviewing and analysing psychologists' response. A quantitative comparison where participants using a riker scale to score their preference between models is valid (which we did in a different study), but not our focus here.  We also think that if we compare our models with non-interpretable but simple models will unfairly cast favor on our model and interfere the interviewee's perspectives.
>
> The main goal of the qualitative study is to carry out a tailored model demo for a highly specialized group, and understand their need to build trust in XAI. The goal is not to our model is absolutely superior in terms of numbers and the only way to go forward for mental health diagnosis using AI. Our model and the paper itself serves as an example of interpretable AI where practitioners can make informed decisions (whether they use it or not and how they use it).

---

> > ### Comment · Reviewer_1eMZ · 2024-11-20
> > **Re: author response**
> >
> > Thanks to the authors for the additional clarification!
> >
> > > Regarding global reweighting and model accuracy
> >
> > Thanks to the authors for the response! However, I'm concerned about the reliability of conclusions made about interpretability when the model performance / predictive power is so low. Was the overall performance stated during the qualitative study?
> >
> > > Regarding dataset set
> >
> > While the DAIC-WOZ data is video/audio/text based, several prior works have extracted tabular, intepretable features and used conventional ML models for training [1]. That is, the features need not be just the component scores. Results (atleast in terms of accuracy - interpretability tradeoffs) on such a dataset would strengthen the generalizability of findings.
> >
> > Given that these limitations are unaddressed, I'll keep my score unchanged for now.
> >
> > *References*
> >
> > [1] Villatoro-Tello, E., Ramírez-de-la-Rosa, G., Gática-Pérez, D., Magimai.-Doss, M., & Jiménez-Salazar, H. (2021, October). Approximating the mental lexicon from clinical interviews as a support tool for depression detection. In Proceedings of the 2021 international conference on multimodal interaction (pp. 557-566).

---

### Meta-Review · Area_Chair_KDei · 2024-12-21

**Metareview:**

The paper studies interpretability and adaptability of a NN-KNN model in the context of mental disorder diagnosis. Evaluation is made through a qualitative study.


Strengths:
 - Qualitative study of method reflects the end-user of the product

Weaknesses
 - The problem is not well-motivated. It is unclear if the paper is trying to advocate for this specific model or study explanable AI principles overall. Model performance overall is bad, making it unclear why understanding this particular model is so important.
 - Several seemingly arbitrary choices (e.g., global feature reweighting) which may affect model performance
 - Questionnaire focuses on tunable weights instead of overall explanability as initially described.
 - No baselines, e.g., a more simple LR model, to compare again.
 - Title of "all models are biased ..." is not reflective of the actual paper contents
 - Hard to understand what the ICLR community would gain from this work.

The paper as written appears overly specific to one model, one use case, and one evaluation method. It is difficult to determine what the ICLR community would gain from this work. As a result, I am recommending reject.

**Additional Comments On Reviewer Discussion:**

Much of the discussion between the reviewers and the authors focused on justifying limitations of the study. Examples included why the training data was so small, why the model accuracy was so bad, and why the title seemed to not match the actual contributions of the paper. Additionally, the paper authors petitioned the Area Chair (myself) directly because they felt that the reviewers did not fully appreciate their work, particularly the qualitative study.

Although fields like psychology regularly rely on qualitative study, ICLR and the ML community value and often expect quantitative evaluation --- or at least models with high performance compared to baseline results. As I explained to the paper authors, the reviewers' emphasis on and requests for quantitative rigor are appropriate for this venue.

This paper lacks several key components, which weighed heavily in my recommendation.

---

### Decision · Program_Chairs · 2025-01-22

Reject